# Effect of virtual reality-simulated exercise on sympathovagal balance

Sheaza Ahmed[1], Maryam Safdar[1], Courtney Morton[2], Nicolette Soave[2], Riya Patel[2], Kenia Castillo[1], Sophie Lalande[3], Linda Jimenez[2], Jason H. Mateika[1,4], Robert Wessells[1]*

1 Department of Physiology, Wayne State University, Detroit, Michigan, United States of America,
2 Department of Kinesiology, Health and Sport Studies, Wayne State University, Detroit, Michigan, United States of America, 3 Department of Kinesiology and Health Education, The University of Texas at Austin, Austin, Texas, United States of America, 4 Department of Internal Medicine, Wayne State University, Detroit, Michigan, United States of America

* rwessell@med.wayne.edu

**Data Availability Statement:** All relevant data are within the manuscript and its Supporting Information files.

**Funding:** This work was supported by a Jack Ryan Pilot Award from the Wayne State School of

## Abstract

Discovery of therapeutic avenues to provide the benefits of exercise to patients with enforced sedentary behavior patterns would be of transformative importance to health care. Work in model organisms has demonstrated that benefits of exercise can be provided to stationary animals by daily intermittent stimulation of adrenergic signaling. Here, we examine as a proof of principle whether exposure of human participants to virtual reality (VR) simulation of exercise can alter sympathovagal balance in stationary humans. In this study, 24 participants performed 15 minutes of cycling exercise at standardized resistance, then repeated the exercise with a virtual reality helmet that provided an immersive environment. On a separate day, they each controlled a virtual environment for 15 minutes to simulate exercise without actual cycling exercise. Response to each treatment was assessed by measuring heart rate (HR), norepinephrine, and heart rate variability, and each participant's response to virtual exercise was compared internally to his/her response to the actual cycling. We found that neither post-exercise norepinephrine nor post-exercise HR was significantly increased by VR simulation. However, heart rate variability measured during virtual exercise was comparable to actual cycling in participants that engaged in moderate exercise, but not in those that engaged in high-intensity exercise. These findings suggest that virtual exercise has the potential to mimic some effects of moderate exercise. Further work will be needed to examine the longitudinal effects of chronic exposure to VR-simulated exercise.

## Introduction

Chronic aerobic exercise acutely improves health and lowers the long-term probability of cardiovascular disease, diabetes, obesity, cancer (colon and breast), osteoporosis, hypertension and lipid abnormalities [1–6]. Moreover, chronic exercise has been shown to enhance glucose homeostasis, insulin sensitivity, and body composition [7–9], while lowering anxiety,

Medicine and Department of Kinesiology, and by
NIH/NIA R21 AG055712 and RO1AG059683 to
RJW. JHM was supported by awards
(R01HL085537, R56HL142757) from the National
Heart, Lung and Blood Institutes. KC, NS, CM and
MS were supported by the Wayne State Physiology
SURF program. The funders had no role in study
design, data collection and analysis, decision to
publish, or preparation of the manuscript.

**Competing interests:** The authors have declared
that no competing interests exist.

depression and stress [10–12]. Despite these diverse benefits, only 53% of the United States population exercise regularly [13]. This lack of exercise may result from lifestyle preferences, job requirements, or immobility associated with hospitalization or chronic injury. Partially because of this inactivity, 40% of the population is obese and suffers from diabetes (15%) and/ or cardiovascular disease (12.1%) [13]. Therefore, alternative strategies that promote outcomes typically linked to exercise are required.

One possible strategy is to employ VR technology to serve as an alternative treatment in combination with, or as a replacement for, bona fide physical exercise. VR is an emerging technology that creates an interactive virtual environment, incorporating auditory and visual simulations, that provide an immersive experience [14]. The perception of reality is achieved by the use of head-mounted displays and touch-sensitive gloves or controllers [15]. Movements of the head and body are tracked and correspond to the virtual movement, which allows the user to explore and be immersed in the virtual environment [16]. Immersing participants in a virtual environment leads to a sensation of agency, in which participants feel a sense of identification and control over the avatar [17]. VR technology has already been used in the treatment of psychiatric disorders, pain, physical disabilities and child behavior [15, 18–20], and to augment procedures associated with plastic surgery and dentistry [21, 22].

Chronic application of simulated exercise can induce beneficial outcomes in stationary *Drosophila* [23]. In flies, octopamine, the invertebrate equivalent of norepinephrine, is critical for exercise adaptation. Intermittent activation of octopaminergic neurons through genetic means is sufficient to drive beneficial exercise adaptions in stationary *Drosophila* [23] that are indistinguishable from those seen in flies that actually performed daily exercise.

The findings in *Drosophila*, coupled with the fact that epinephrine and norepinephrine increase during exercise in humans, and the fact that VR has been shown to be capable of increasing HR in stationary users [24, 25], led us to propose that intermittent stimulation of adrenergic signaling might have therapeutic benefit in humans with a sedentary lifestyle [26].

As a first step toward this goal, we hypothesized that simulated exercise may lead to increases in sympatho-vagal balance and decreases in parasympathetic nervous system activity that are ultimately coupled to increased norepinephrine release and heart rate. To test this hypothesis, we measured heart rate variability during virtual exercise and norepinephrine levels before and immediately after virtual exercise, and compared them to the same parameters measured during high or moderate-intensity exercise. The goal of this study was to establish the practicality of utilizing VR as a method of intermittent sympathetic stimulation, as a precursor to further studies examining the effects of such treatment over a longer-term on healthy participants or patients with chronic disease or disability.

## Methods

### Participants

Twenty-four healthy individuals (S1 and S2 Tables) were recruited by advertising on Wayne State University's website and through departmental listservs as a convenience sampling strategy of healthy adults able to tolerate VR. Following recruitment, participants attended a pre-screening interview to review the study protocol and to determine if the participants met the inclusion/exclusion criteria. The inclusion criteria included an absence of cardiovascular disease and diabetes, the absence of medications that could affect the response to exercise, such as beta-blockers or selective serotonin reuptake inhibitors (SSRI), and the absence of anxiety or nausea if participants had previous experience with VR.

## Ethics

The Wayne State University Institutional Review Board approved the study and all participants provided written consent before participating in the study.

## Study protocol

The twenty-four participants were allocated to an experimental group or a no prior exposure control group (Fig 1). The experimental group was comprised of 17 participants (11 men and 6 women) who completed 3 sessions with varying stimuli. Each participant performed the 3 sessions in the same order and each session was separated by a week. The no prior exposure group was comprised of 7 participants (4 men and 3 women) who completed a single VR-simulated exercise session. Participants were asked to abstain from caffeinated beverages and nicotine for at least 2 hours before a session. For the initial session, each participant in the experimental group performed 15 minutes of exercise on a recumbent bicycle (EX–Fig 1). The participants pedaled at a frequency of 72–94 RPM that resulted in a speed of 9–10 MPH with added resistance set to mimic the exercise intensity of the pre-set HOLOFIT software (see *Equipment* for description of software). During the second session (EX + VR—Fig 1), each participant performed 15 minutes of exercise while wearing a VR headset that provided visual and auditory sensory inputs. Speed and movement of the avatar in the virtual reality environment was controlled by the participant using a recumbent bicycle. In the VR headset, when participants looked down they could see the avatar's legs matching the speed and the movement of their own legs on the recumbent bicycle. Resistance and speed of cycling was identical to the EX session.

Throughout the third session, the participants wore the VR headset and instead of performing physical exercise for 15 minutes, the participants controlled the VR environment with a keyboard (VR—Fig 1). Consequently, participants were able to control the virtual environment and receive visual input consistent with exercise, but did not perform aerobic exercise. The no prior exposure group also completed 15 minutes of virtual exercise. However, in this case, the session was completed without prior exposure to VR or the recumbent bicycle (VR NPE–Fig 1). The no prior exposure group served as a control for the effects of learned expectations or familiarity with the environment.

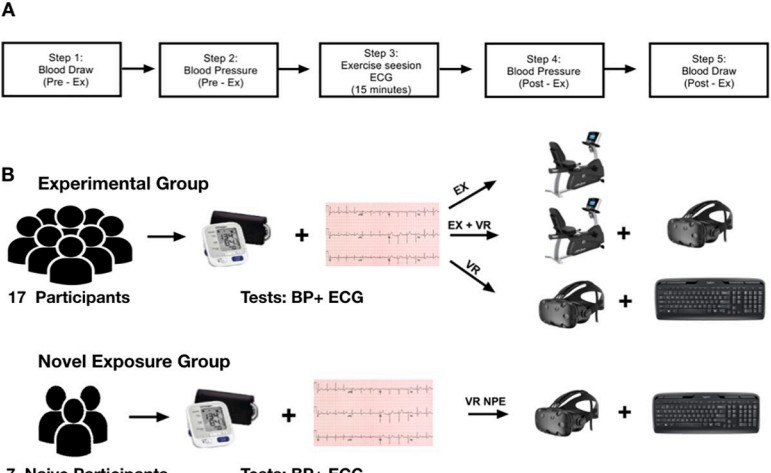

**Fig 1. Experimental setup.** A, Step-wise description of the exercise protocol; B, Details of the experimental and the no prior exposure group.

For all groups, HR, blood pressure and plasma norepinephrine levels were measured immediately before and after the exercise. HR was also monitored throughout the exercise period via electrocardiogram. Exercise intensity during EX was determined by dividing the average HR over the last five minutes of exercise by the maximum HR calculated using the equation:$191.5 - (0.007 \times age^2) = $ Maximum HR [27]. Participants were separated into high-intensity or moderate-intensity groups on a *post-hoc* basis, according to their measured HR during the initial cycling session, using a cutoff of 70% maximum HR.

## Equipment

A VR headset (HTV Vive, Taiwan) along with a gaming mini-PC (Zotac ZBOX MAGNUS EN1070K, Dongguan City, China) was used. The virtual environment was created using HOLOFIT commercial software (HOLODIA, Zurich, Switzerland) and provided participants with visual stimuli of cycling through a desert canyon environment. The software was altered so that a keyboard or a recumbent bicycle could be used to control the virtual environment. The keyboard was used because of the available compatibility with HOLOFIT software and the requirement of minimum movement of participants. A recumbent Lifecycle bike (Life Fitness activate series OSR-XX, Illinois, USA) with cSAFE protocol was used since it was compatible with the HOLOFIT commercial software.

## Data acquisition and analysis

Two ml of venous blood was obtained from the antecubital vein before and after each exercise session. The sampled blood was centrifuged and serum was aliquoted and stored at -80˚C for subsequent analyses. Serum concentrations of norepinephrine levels were assessed in duplicates using an enzyme-linked immunosorbent assay (Abnova, Taiwan). Briefly, following sample preparation, extraction and acylation, an enzyme solution and serum samples were added into wells of a microplate plate. Samples were incubated for 30 min before adding antiserum into all wells and incubating for 2 hours. The plate was then washed, an enzyme conjugate was added into all wells, and samples were incubated for 30 min. The plate was washed again with wash buffer, substrate was added into all wells and incubated for approximately 25 min. The stop solution was added to the plate and absorbance was read within 10 min using a microplate reader set to 450 nm.

Blood pressure was measured before and after each exercise session, using a blood pressure monitor (Omron 5 Series, Omron, Lake Forest, IL). Intra-correlation coefficients within each group (Groups HI and MI are described below) were greater than or equal to 0.86 for systolic and 0.75 for diastolic, indicating a high degree of reproducibility between baseline measurements for each individual. In addition, an electrocardiogram was recorded throughout (i.e. 15 minutes) the physical and virtual exercise sessions using a two-lead configuration. Data acquisition was performed using LabChart Software V8 (ADInstruments, Sydney, Australia).

Average measures of blood pressure, norepinephrine and HR, are shown for each participant before and after each exercise session in S1 Table, with individual raw data for the same measures shown in S2 Table. These parameters were not significantly different between men and women. Thus, data were combined for analysis. Although there was substantial variation in baseline norepinephrine measurements (S1 Table), we found no correlation between baseline norepinephrine and the degree of change in norepinephrine after exercise (S1 Fig), except in the VR only group, where a low baseline norepinephrine significantly correlated with a high change in norepinephrine (VR: P = 0.001, r = 0.74, CI = -0.92 to -0.30).

Based on the increase in HR (i.e., % of maximum HR) during the initial exercise session (EX), participants were divided into two distinct groups that exercised at high ($<$ 70% maximum HR) or moderate intensity ($>$ 70% maximum HR) (HI and MI, respectively). This cutoff

was chosen because it best described the data from this set of participants and is similar to the target HRs used by the American Heart Association to delineate high and moderate intensity [28]. The HI group was comprised of 6 participants, while the MI group was comprised of 11 participants.

Spectral analysis of heart rate variability was determined from the electrocardiogram recorded during the last 5 minutes of each exercise session as previously described [29]. The last 5 minutes in each exercise session were chosen because the participants were expected to undergo maximum exertion during that time. Spectral components for heart rate variability were measured in absolute units ($ms^2$/Hz). Power in the low-frequency range (LF R-R: 0.04 to 0.15 Hz), and high-frequency range (HF R-R: 0.15 to 0.40 Hz) was determined and the LF/HF ratio was calculated. Modifications in the high-frequency range was considered to represent changes in parasympathetic nervous system activity while the LF/HF ratio was considered to reflect sympathovagal balance [30]. The heart rate variability data was ln transformed before statistical analysis was completed because the data was not normally distributed. HF and LF/HF were calculated using the data collected during the initial exercise session (EX), the virtual exercise session (VR) and the virtual exercise session with no prior exposure (VR NPE).

### Statistical analysis

Absolute measures of blood pressure, norepinephrine and HR, before and after exercise, were compared across sessions. Results were analyzed by a two-way repeated measures ANOVA with a Tukey post-hoc test. The factors in the design were time (before vs. after exercise) and session (EX vs. EX+VR vs. VR). In addition, the data collected before and after VR were compared between the experimental group and no prior exposure group (i.e. VR NPE), using a two-way ANOVA followed by a Tukey post-hoc test. The factors in the design were time point (before vs. after exercise) and group (VR vs. VR NPE). For all comparisons, p-values of 0.05 or lower were considered significant. Statistical analysis was performed with GraphPad Prism 7 (Graph Pad, La Jolla, CA), except for intra-class correlation (ICC) and effect size (ES) analysis, which was performed with SPSS v26.0 (IBM Corporation, Armonk, NY).

A Pearson correlation was used to determine the association between % maximum HR achieved during the last five minutes of exercise in the initial session (EX) and the difference in HR and heart rate variability during EX and VR (e.g. HR during the last 5 minutes of EX– HR during the last 5 minutes of VR), or norepinephrine immediately after EX and VR.

Lastly, an unpaired t-test was used to compare HR and heart rate variability measures during the last five minutes of EX and VR between groups group, or norepinephrine levels immediately following EX or VR, in the HI and MI groups.

## Results

### Blood pressure, heart rate and norepinephrine

Systolic (Fig 2A) and diastolic (Fig 2B) blood pressure was not significantly different immediately after any treatment. HR immediately following EX (P < 0.0001, ES = 1.9, CI = -40.21 to 16.03) and EX + VR (P < 0.0001, ES = 1.9, CI = -37.76 to -22.09) was significantly elevated compared to baseline but this was not the case following VR. Norepinephrine measured immediately after exercise was not greater than baseline for any pairwise comparison.

### Heart rate variability

To compare the acute effects of physical and VR exercise, we analyzed electrocardiogram traces from the last five minutes of EX and VR to determine HR and heart rate variability for

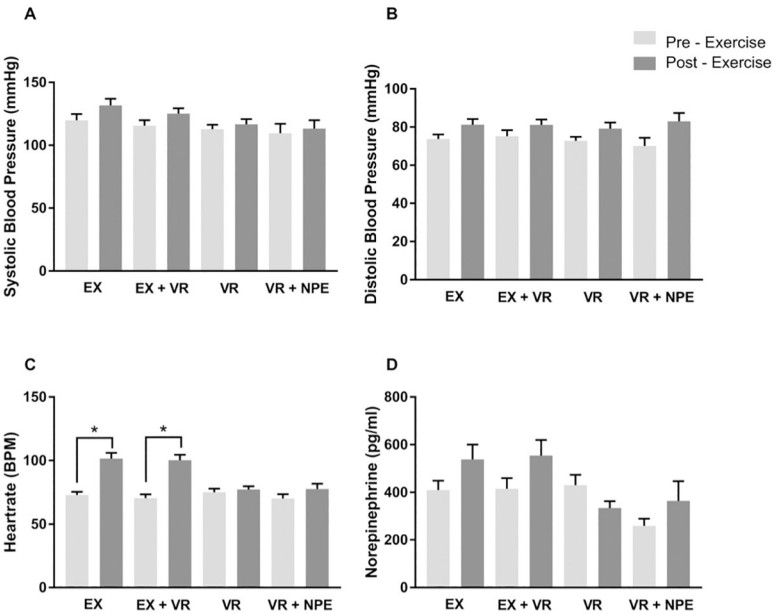

**Fig 2. Parameters measured before and after the exercise sessions.** A, Systolic blood pressure; B, Diastolic blood pressure; C, HR and D, Norepinephrine. The main effect of the treatment was significant across all groups for systolic blood pressure, HR and norepinephrine, but there were no significant differences in pairwise comparisons for individual groups. HR was significantly increased in the exercised group and in EX+VR group. Abbreviations: EX, Physical exercise group; EX+VR, Physical exercise group with a VR headset on; VR, Virtual exercise group; VR NPE, Virtual exercise group with no prior exposure to physical and virtual exercise.

each participant. We then plotted the difference in HR during EX and VR for each participant. In most participants, HR during EX was higher than HR during VR (Fig 3A).

The intensity of physical exercise (as indicated by the % of maximum HR achieved during the last 5 minutes of exercise during EX) predicted the difference in HR during EX and VR for an individual participant (Fig 3A). Specifically, participants that engaged in high-intensity physical exercise during EX experienced a greater increase in HR (Fig 3A), norepinephrine (Fig 3B) and sympathovagal balance (Fig 3D) compared to VR exercise. In contrast, a greater decrease in parasympathetic nervous system activity (lnHF) was evident during EX compared to VR in those participants engaged in high intensity exercise (Fig 3C). A similar physiological response during EX and VR was evident in those participants that engaged in moderate-intensity exercise during EX. The point on the correlation line where the response to actual and virtual exercise would be identical is marked with a black arrow (Fig 3).

Those participants whose HR was greater than 70% of maximum HR during EX were classified as high intensity (HI) exercisers while the participants whose HR was less than 70% of maximum were considered to have engaged in moderate intensity (MI) exercise. This grouping resulted in 6 participants who engaged in high-intensity exercise (80.9 ± 2.8% maximum HR) and 10 that engaged in moderate-intensity exercise (60.0 ± 2.7% maximum HR). Data from one participant was not available because of technical issues.

HR during EX and VR was significantly higher in the HI compared to the MI group (EX: P < 0.0002, ES = 2.7, CI = 22.4–55.8 & VR: P < 0.04, ES = 1.1, CI = 0.9–23.0). In addition, the HR difference between EX and VR was greater in magnitude in the HI compared to the MI group (P < 0.02, ES = 1.4, CI = 5.9–48.7) (Fig 4A). Post-exercise norepinephrine was greater after EX in the HI compared to the MI group (p < 0.02, ES = 1.3, CI = 52.7–520.3). However, post-exercise norepinephrine following VR was similar in the HI and MI group (p = 0.36,

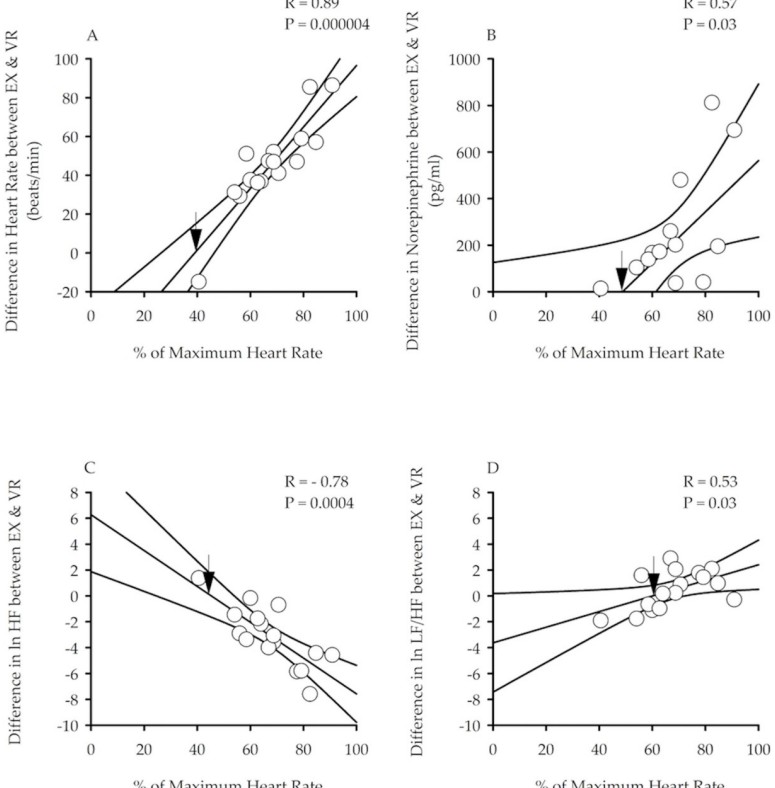

**Fig 3. Correlations between % maximum heart rate and other parameters.** Percent of maximum HR measured during exercise is plotted against differences in HR (A), norepinephrine (B), parasympathetic nervous system activity (ln HF) (C) and sympathovagal balance (ln LF/HF) (D) measured from the last 5 minutes of actual exercise (EX) and virtual exercise (VR). Measurements in A, C, and D are taken during exercise, while measurements in B were taken immediately after exercise. Note that the difference in HR, norepinephrine and sympathovagal balance measured during EX compared to VR was greater in those participants that exercised at a higher intensity during EX, as determined by the percentage of maximum HR. Likewise, a greater decrease in parasympathetic activity during EX compared to VR was evident in those participants that exercised at a higher intensity. Lastly, note that the black arrows in each graph represent the point on the predicted line where the response to actual and virtual exercise would be similar. These points indicate that responses during VR exercise are capable of inducing physiological responses similar to that induced during actual mild to moderate exercise (40% - 60% of maximum HR). 95% confidence intervals are shown on each scatterplot.

ES = 0.5, CI = - 84.4–215.7). Parasympathetic nervous system activity was reduced during EX in the HI compared to the MI group (P < 0.03, ES = 1.0, CI = - 4.1–0.09) (Fig 4C). In contrast, parasympathetic nervous system activity was similar during VR in the HI and MI group (p = 0.24, ES = 0.7, CI = - 0.5–1.9). Sympathovagal balance was similar during EX in the HI and MI group (p = 0.73, ES = 0.2, CI = - 1.0–1.4). In contrast, sympathovagal balance was reduced during EX in the HI compared to the MI group (P < 0.03, ES = 1.2, CI = - 1.7 –- 0.1) (Fig 4D).

## Discussion

VR has found multiple effective therapeutic uses in recent years, to increase exercise motivation [31], and to stimulate HR in stationary users [24, 25]. Its use to encourage patients to complete therapeutic recovery regimens, sometimes called exergaming, is particularly well-developed [32, 33]. Movement based VR exer-gaming is also becoming a popular exercise modality in healthy adults, and has been shown to increase HR and metabolic rate (i.e. oxygen

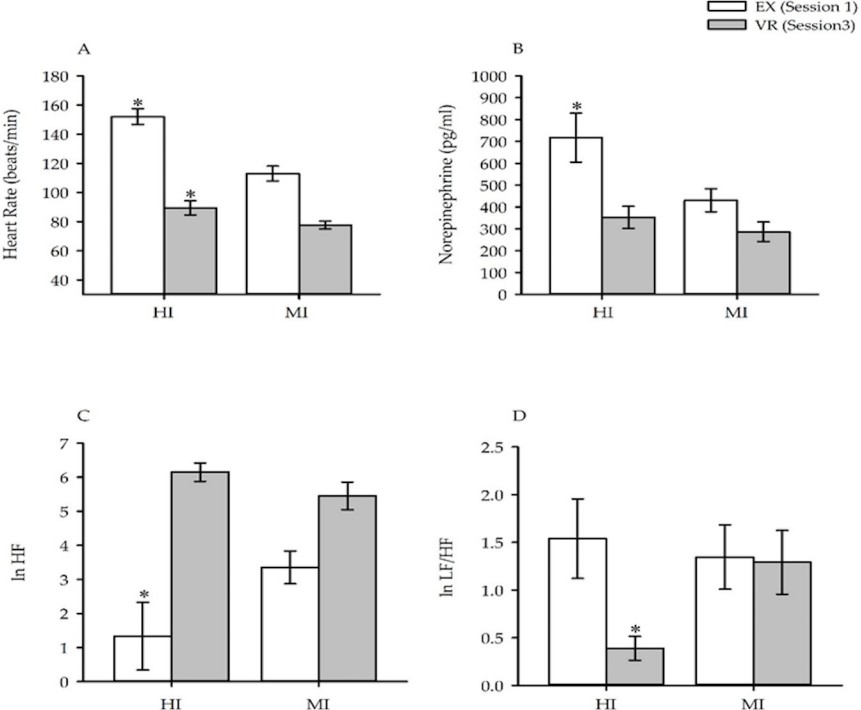

**Fig 4. Parameters measured for the HI and MI exercise groups during EX and VR sessions.** A, HR measured during the last five minutes of exercise; B, Norepinephrine measured after the exercise sessions; C, Parasympathetic nervous system activity (ln HF) measured during the last five minutes of exercise; D, Sympathovagal balance (ln LF/HF) measured during the last five minutes of exercise. Note that sympathovagal balance (ln LF/HF) was greater during VR in the MI compared to the HI group and that the response in the MI group was similar to the response measured from this group during EX. Abbreviations: EX—Physical exercise; VR—Virtual exercise; HI—High intensity; MI—Moderate intensity. *—significantly different from MI.

consumption) from 'moderate' to 'vigorous' levels depending on the VR experience [34]. However, the use of VR to induce a physiological response that mimics exercise is not as well studied as its role in promoting exercise motivation. This study set out to address the question of whether VR exercise could increase HR and norepinephrine release to a comparable extent as actual cycling for the same duration.

Although others have reported increased HR during VR when participants caused an avatar to move [24], we did not observe consistent significant effects of VR on HR across all participants, although there was substantial individual variance. During the course of measuring the baseline exercise response, we noticed that participants could easily be divided into two groups based on their HR response to actual cycling (i.e. HI and MI groups). This division likely reflects differences in baseline fitness in our participants, which is consistent with results from our observations of the participants. Even though the rate and resistance of the bicycle were identical between participants, some participants clearly found the set point of exercise more difficult than others.

Virtual exercise was not able to replicate the physiological responses measured during HI exercise. However, among participants experiencing MI exercise conditions, norepinephrine levels and measures of heart rate variability were not statistically different from those measured during VR. This finding suggests that virtual exercise has the potential to successfully replicate moderate, exercise-induced changes to sympathovagal balance, but is unlikely to be effective at replicating intense endurance exercise.

There were certain limitations to this study. The study participants had a wide variety of physical fitness, including some, presumably at a high baseline fitness level, that showed minimal increases in HR and norepinephrine during both actual and virtual exercises, lowering the overall average responses. Other participants had unusually high baseline norepinephrine levels, for unknown reasons, which tended to blunt their exercise-induced change in norepinephrine levels. Participants were asked to refrain from stimulants for at least two hours before coming to the lab, and this may not have been sufficient to prevent effects of caffeine or nicotine. A few participants were apprehensive about the blood draw, which might have affected their HR and norepinephrine measurements. Although we made sure to take the post-exercise blood draws immediately after the exercise sessions, even a slight delay could have affected the change in norepinephrine levels compared to the baseline, since the half-life of norepinephrine in blood is 2–2.5 minutes [35]. The variation in baseline HR amongst participants was also high, for unknown reasons.

In summary, these results offer an early indication that virtual exercise may be able to mimic some of the effects on the autonomic nervous system associated with moderate exercise, though not highly strenuous endurance exercise. This possibility merits further investigation, as moderate exercise has profound salutary effects on stationary patients. Mild/Moderate exercise has been shown to lower blood pressure [36, 37], lower total serum cholesterol [38], maintain a healthy LDL/HDL ratio [38], improve body composition [38], protect myocardial function [39], prevent glucose intolerance [40], improve autonomic nervous system [41] and increase insulin sensitivity [42]. Patients with extended debilitating illnesses or injuries that lack access to daily exercise often suffer reduced metabolic fitness, and exposure to the benefits of moderate exercise could preserve such patients against compounding factors caused by sedentary periods. Given the low cost and lack of side effects of virtual exercise treatment, it has the potential, once better understood, to become an effective treatment tool for a wide cross-section of sedentary patients. This study provides proof-of-principle for more extensive and detailed future studies to further develop this potential treatment.

## Supporting information

**S1 Fig. Correlations between Δ norepinephrine and baseline norepinephrine.** A, Correlation in cycling group; B, Correlation for group that cycled using the VR environment; C, Correlation for group that experienced virtual exercise without cycling; D, Correlation for group that experienced vitual exercise without prior exposure.
(TIF)

**S1 Table. Participants' characteristics, hemodynamic variables and norepinephrine levels pre- and post-exercise as mean +/- SD.**
(DOCX)

**S2 Table. Participants' characteristics, hemodynamic variables and norepinephrine levels pre- and post-exercise.**
(DOCX)

## Acknowledgments

We are grateful for the help of Alyson Sujkowski and Charles Chung with figures and statistical analysis. We would like to acknowledge Keith Myszenski and Theodore Eisenstein for helping with the VR setup, and Holodia for the modified software used in our study.

## Author Contributions

**Conceptualization:** Sophie Lalande, Linda Jimenez, Jason H. Mateika, Robert Wessells.

**Data curation:** Sheaza Ahmed, Maryam Safdar, Courtney Morton, Nicolette Soave, Riya Patel, Kenia Castillo, Sophie Lalande, Linda Jimenez, Jason H. Mateika, Robert Wessells.

**Formal analysis:** Sheaza Ahmed, Kenia Castillo, Sophie Lalande, Jason H. Mateika.

**Funding acquisition:** Jason H. Mateika, Robert Wessells.

**Investigation:** Sheaza Ahmed, Maryam Safdar, Courtney Morton, Nicolette Soave, Riya Patel, Linda Jimenez.

**Methodology:** Sheaza Ahmed, Maryam Safdar, Courtney Morton, Nicolette Soave, Riya Patel, Sophie Lalande, Jason H. Mateika, Robert Wessells.

**Project administration:** Jason H. Mateika, Robert Wessells.

**Resources:** Robert Wessells.

**Supervision:** Sheaza Ahmed.

**Writing – original draft:** Sheaza Ahmed, Robert Wessells.

**Writing – review & editing:** Sophie Lalande, Linda Jimenez, Jason H. Mateika.

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
