## [Decision Letter · Decision Letter 0]

27 Apr 2020

PONE-D-20-08695

Effect of Virtual Reality-Simulated Exercise on Sympathetic Nervous System Activity

PLOS ONE

Dear Dr Wessells,

Thank you for submitting your manuscript to PLOS ONE. After careful consideration, we feel that it has merit but does not fully meet PLOS ONE’s publication criteria as it currently stands. Therefore, we invite you to submit a revised version of the manuscript that addresses the points raised during the review process. In addition, please be cautious about the interpretations of heart rate variability measures for sympathetic modulations since it becomes more controversial with recent evidence.

We would appreciate receiving your revised manuscript by Jun 11 2020 11:59PM. To enhance the reproducibility of your results, we recommend that if applicable you deposit your laboratory protocols in protocols.io, where a protocol can be assigned its own identifier (DOI) such that it can be cited independently in the future. For instructions see: http://journals.plos.org/plosone/s/submission-guidelines#loc-laboratory-protocols

We look forward to receiving your revised manuscript.

Kind regards,

Huimin Yan

Academic Editor

PLOS ONE

Journal Requirements:

Reviewers' comments:

Reviewer's Responses to Questions

**Comments to the Author**

1. Is the manuscript technically sound, and do the data support the conclusions?

Reviewer #1: Partly

Reviewer #2: Yes

2. Has the statistical analysis been performed appropriately and rigorously? 

Reviewer #1: Yes

Reviewer #2: Yes

3. Have the authors made all data underlying the findings in their manuscript fully available?

Reviewer #1: Yes

Reviewer #2: Yes

4. Is the manuscript presented in an intelligible fashion and written in standard English?

Reviewer #1: Yes

Reviewer #2: Yes

5. Review Comments to the Author

Reviewer #1: It was with great excitement that I was asked to review this manuscript. The notion of using VR as an exercise modality is novel and the data are just emerging. This manuscript contributes to what little we know about VR exercise.The data suggested that moderate exercise completed using VR had similar responses to moderate intensity cycling, but not not high-intensity cycling. While I feel that the upside for this manuscript is very great, there are certain nuances that reduced my enthusiasm.

Major issues:

-In the abstract and even in the manuscript the authors mention sympathetic activity measured by heart rate variability. In the methods when the authors outline the physiological mechanisms behind the HRV measurements, none of them are listed as having SNS influence. The lone one would be LFnu, but in reality that is nothing more than 1-HFnu so if anything it is a vicarious measure of SNS outflow. If this is the case, then how do the authors justify HRV as measures of SNS outflow? On page 14, line 249 the authors state that LF/HF is a measure of SNS outflow, but that is not what they said on page 12, lines 188-89. This makes this paper really hard to follow, and confusing.

-No hypothesis is provided. Without this vital bit of information it is impossible to know what the authors think will happen with the data in response to the intervention.

Minor issues:

-In the abstract, line 39, I read the word, 'caused.' I'd be cautious with this as the data you are presenting is nothing more than a correlate.

-Page 3, first paragraph. From a writing perspective I would always say that a paragraph is 3 sentences, this one is two.

-Table 1 is very hard to follow. A lot of abbreviations and no caption. Based on its placement none of these things MI, HI or NPE have been introduced. Also, none of the variables have units.

-Page 9, line 136. This method for calculating max HR needs a reference.

-Page 10, line 159. What is the ICC for BP? No mention of reliability or validity.

-Page 12. No mention of what tests were run for normality, or what data were transformed. This limits the readers comprehension severely.

-RESults. Why was teh decision made to only include p values? This is very limiting as that only addresses type I error, correct? Consider adding ES, CIs, etc, as measures of power.

-References are often incomplete (some are missing the journal name) and they are inconsistent (journal name either missing caps, or is abbreviated).

Reviewer #2: This article examined effects of VR simulation of exercise on sympathetic activity in stationary humans (vs VR exercise cycling). Main findings were that heart rate variability increase with VR exercise simulation, but there was no change in norepinephrine or heart rate with VR simulated exercise. The data is well presented, and paper well written. See minor comments below:

Abstract:

Line 29: Please abbreviate 'virtual reality' as "VR" starting here, and throughout the paper: "... to virtual reality (VR) simulation of exercise..."

Line 31: Change the word "bicycling" to "cycling" throughout, as this is a stationary cycle ergometer (not a true bicycle).

Line 33: Make it clear here that the "simulation" session included no actual exercise: "...to simulate exercise (without cycling exercise)..."

Line: 34: Abbreviate 'heart rate' as "HR" starting here, and throughout the paper.

Line 36: "find" should be "found".

Introduction:

Line 48: Please specify this is "Chronic aerobic exercise" in the first sentence and combine the 1st and 2nd paragraphs .

Line 53: Make it clear that these data are from the US: "...of the population of the United States (US) exercises regularly...".

Line 58: Again, abbreviate 'virtual reality' as "VR" starting here, and throughout the paper.

Line 62: Specify that it is either a "head-mounted display or wall projector".

Line 86: Include this group at the end of the intro: "... with chronic disease or disability."

Methods:

Table 1: Please report this table with individual data as a supplement table. The data for each participant is not necessary here. Instead please re-create this table with 'mean +/- SD in each group, session, and variable pre- and post- exercise (be sure to identify which session was which, and identify abbreviations in the heading (HI, MI, NPE).

Line 111: I'm note sure that 2 hours without stimulants is long enough to clear from a human system. Please state why 2 hours was chosen as a cutoff for abstaining from caffeine and nicotine, and not longer. Also, where any other controls taking into account? Did the participants refrain from exercise prior to the trials? This could have affected baseline norepinephrine levels.

Line 136: Please cite the source of this MaxHR equation.

Line 154: Briefly describe your ELISA protocol (or cite source of similar protocol from your lab).

Line 182: Space out "minutesin".

Line 209: State your alpha-level for significance and any statistical analysis software used.

Discussion:

Line 284: In the discussion, I think it's also important to note that VR exer-gaming is also being used by healthy people for exercise, and has been shown to increase metabolic rate (measured by VO2 consumption) and HR. Metabolic rate would be a good measure for future studies on virtual exercise... Consider including a sentence like this:

"...particularly well-developed [32, 33]. Movement based VR exer-gaming is also becoming a popular exercise modality in health adults, and has been shown in increase HR and metabolic rate (oxygen consumption, VO2) from 'moderate' to 'vigorous' levels depending on the VR experience (Cite Gomez, 2018). However, the use of VR to induce...". Please cite Gomez 2018 here (https://pubmed.ncbi.nlm.nih.gov/30325233/).

References:

Please ensure that references are correctly formatted (some journal titles are capitalized, others are not).

6. PLOS authors have the option to publish the peer review history of their article (what does this mean?). If published, this will include your full peer review and any attached files.

Reviewer #1: No

Reviewer #2: Yes: James R. Bagley

---

## [Author Response · Author response to Decision Letter 0]

11 May 2020

We have attached a full pint by point letter responding to each reviewer comment. The major concern about HRV interpretation has been addressed by modifying our description of results to focus on sympathovagal balance rather than sympathetic activity per se. This change is reflected in the revised manuscript title. See letter for other responses. We thank the reviewers for their thoughtful comments,

---

## [Decision Letter · Decision Letter 1]

9 Jun 2020

PONE-D-20-08695R1

Effect of Virtual Reality-Simulated Exercise on Sympathovagal Balance

PLOS ONE

Dear Dr. Wessells,

Thank you for submitting your manuscript to PLOS ONE. After careful consideration, we feel that it has merit but does not fully meet PLOS ONE’s publication criteria as it currently stands. Therefore, we invite you to submit a revised version of the manuscript that addresses the points raised during the review process.

We look forward to receiving your revised manuscript.

Kind regards,

Huimin Yan

Academic Editor

PLOS ONE

Reviewers' comments:

Reviewer's Responses to Questions

6. Review Comments to the Author

Reviewer #1: This revised manuscript is a significant improvement. I do have a few suggestions that I think will further improve this manuscript.

-Page 4, line 78. The authors simply state, '...modifications in autonomic control...' as part of their hypothesis. This is pretty non-descriptive and doesn't really help the reader. I'd suggest being specific here.

-Page 12, line 250. The authors state '(HF)' but don't they really mean (lnHF)?

-Page 12, line 266. Post exercise or post-exercise? The authors keep changing. I'd suggest double checking document for consistency.

Reviewer #2: All of my previous comments/suggestions have been addressed. I feel this paper has been made stronger based on the edits made regarding my, and the other reviewers', comments.

---

## [Author Response · Author response to Decision Letter 1]

12 Jun 2020

Response To Reviewers

We thank the reviewers for their suggestions and criticisms which have substantially improved the manuscript. The reviewer comments are listed below with our responses.

Page 4, line 78. The authors simply state, '...modifications in autonomic control...' as part of their hypothesis. This is pretty non-descriptive and doesn't really help the reader. I'd suggest being specific here.

We changed the sentence to read as follows:

As a first step toward this goal, we hypothesized that simulated exercise may lead to increases in sympatho-vagal balance and decreases in parasympathetic nervous system activity that are ultimately coupled to increased norepinephrine release and heart rate.

Page 12, line 250. The authors state '(HF)' but don't they really mean (lnHF)?

We have changed this to say (lnHF).

Page 12, line 266. Post exercise or post-exercise? The authors keep changing. I'd suggest double checking document for consistency.

We have double-checked and the manuscript now consistently uses hyphens after each use of “post”.

---

## [Decision Letter · Decision Letter 2]

23 Jun 2020

Effect of Virtual Reality-Simulated Exercise on Sympathovagal Balance

PONE-D-20-08695R2

Dear Dr. Wessells,

We’re pleased to inform you that your manuscript has been judged scientifically suitable for publication and will be formally accepted for publication once it meets all outstanding technical requirements.

Kind regards,

Huimin Yan

Academic Editor

PLOS ONE

Additional Editor Comments (optional):

Reviewers' comments:

Reviewer's Responses to Questions

**Comments to the Author**

1. If the authors have adequately addressed your comments raised in a previous round of review and you feel that this manuscript is now acceptable for publication, you may indicate that here to bypass the “Comments to the Author” section, enter your conflict of interest statement in the “Confidential to Editor” section, and submit your "Accept" recommendation.

Reviewer #1: All comments have been addressed

2. Is the manuscript technically sound, and do the data support the conclusions?

Reviewer #1: Yes

3. Has the statistical analysis been performed appropriately and rigorously? 

Reviewer #1: Yes

4. Have the authors made all data underlying the findings in their manuscript fully available?

Reviewer #1: Yes

5. Is the manuscript presented in an intelligible fashion and written in standard English?

Reviewer #1: Yes

6. Review Comments to the Author

Reviewer #1: (No Response)

7. PLOS authors have the option to publish the peer review history of their article (what does this mean?). If published, this will include your full peer review and any attached files.

Reviewer #1: No

---

## [Editor Report · Acceptance letter]

26 Jun 2020

PONE-D-20-08695R2 

Effect of Virtual Reality-Simulated Exercise on Sympathovagal Balance 

Dear Dr. Wessells:

I'm pleased to inform you that your manuscript has been deemed suitable for publication in PLOS ONE. Congratulations! Your manuscript is now with our production department. 

Kind regards, 

on behalf of

Dr. Huimin Yan 

Academic Editor

PLOS ONE